palaeontology

sexual size dimorphism, Strombidae, Kutch, western India, Miocene, fecundity selection

**Author for correspondence:**
Kalyan Halder
e-mail: kalyan.geol@presiuniv.ac.in

# First record of sexual size dimorphism in fossil Strombidae (Mollusca, Gastropoda) from the Miocene of Kutch, western India and its evolutionary implications

## Kalyan Halder and Somnath Paira

Department of Geology, Presidency University, 86/1, College Street, Kolkata 700073, West Bengal, India

KH, 0000-0003-1920-5535; SP, 0000-0002-1631-6101

*Persististrombus deperditus* (Sowerby) from the Lower Miocene of Kutch, Gujarat, western India is represented by two size classes in our collection. Statistical analyses discriminate the size morphs. Large size variations generally result from either (1) sexual differences or (2) ecophenotypic causes. All the living species of the family Strombidae, wherever examined, are characterized by sexual size dimorphism (SSD). *Persististrombus deperditus* shares all the characters of SSD in these recent species. Size variations due to difference in ecological factors generally occur in allopatric populations. Similar variations are known to characterize sympatric sub-populations of molluscs living only in the intertidal zone, where upper and lower shorefaces differ significantly in physico-chemical and biological properties. *Persististrombus deperditus* comes from a stable shelf setting that received less siliciclastic input in response to transgression. Hence, its size dimorphism is considered to have sexual origin. This is the first report of SSD in a fossil strombid gastropod. It is argued that fecundity selection was the primary driving force behind the evolution of SSD in this gonochoristic gastropod species. Hence, the larger morph is the female.

## 1. Introduction

The family Strombidae is present in the Holocene exclusively in shallow tropical and sub-tropical seas. Wherever strombid species are found they are generally quite abundant [1]. The family is characterized by determinate growth. Once the shell

attains the final size it undergoes substantial change in apertural features, such as upturned suture, often flared and thickened outer lip that sometimes bears spines. The final shell size and shape are attained in this family before sexual maturity [2,3].

All the living species of Strombidae are gonochoristic. All the examined species of the family exhibit sexual dimorphism with respect to size where females are larger than males ([4,5] and references therein). Sexual size dimorphism (SSD) in some recent species of Strombidae has been known for a long time (e.g. [6–8]), and Cob *et al*. [9] considered it the general characteristic of the genus 'Strombus'. Size ranges of the two sexes often overlap, but members in the larger size end are generally females, whereas those in the smaller size end are males (e.g. [9]). In spite of the determinate growth and apparent ease of recognition of adulthood because of the presence of adult apertural modifications, sexual dimorphism has never been reported in fossil Strombidae. Dependence of palaeontological studies solely on hard part morphologies and the presence of overlapping size ranges between the sexes might have discouraged such documentation.

Size dimorphism can also result from non-sexual causes. Allopatric populations of a species often exhibit size difference due to difference in ecological factors. Such ecological factors that vary laterally and from shore to offshore with change in depth within a basin are also known to cause size difference in populations of a species. Size difference between populations of recent strombid gastropods that live only a few tens of metres away is known [2].

Here we report size dimorphism in *Persististrombus deperditus* (Sowerby, 1840) from the Miocene of Kutch, western India. We seek to understand the nature of dimorphism and causes of its evolution.

## 2. Dimorphic *Persististrombus deperditus* (Sowerby, 1840)

*Strombus deperditus* Sowerby, 1840 was known from the Lower Miocene of Kutch and Pakistan [10,11]. Harzhauser *et al*. [12] included it (misspelled as *depertitus*) in the genus *Persististrombus* Kronenberg & Lee, 2007 and placed *Strombus nodosus* Sowerby, 1840 into its synonymy. The latter comes from the same stratigraphic level and geographical locality as it. The species is abundant in Kutch. Harzhauser *et al*. [12] provided a formal systematic description of the species.

The species is fusiform, anomphalous, dextrally coiled with elevated and step-like spire. The spire is shorter than the last whorl. The last whorl overlaps about 70% of the preceding whorl. The base is conical and extended into a short and slightly curved anterior siphonal canal. An angular shoulder separates a moderately wide and slightly sloping shelf from the rather flat whorl side. Its surface bears characteristic ornament having prominent spiral ribbing and shoulder tubercles. The species is easily identifiable because of these prominent morphological features. Interestingly, it occurs in a wide range of adult sizes in Kutch. On visual inspection, they can be broadly grouped into two size classes, which do not differ significantly in other features. The number of whorls in all the adult specimens is about eight. The larger and the smaller size groups are referred to here as the macro- and the microconch, respectively (table 1 and figure 1).

The height of the largest macroconch is twice that of the smallest microconch although overlapping is present in size ranges of the two groups. Therefore, it is difficult to assign some specimens to either of the groups. Hence, we have performed statistical analyses to see the viability of these groups.

## 3. Material and methods

### 3.1. Material studied

The Lower Miocene is represented in Kutch, Gujarat by the Khari Nadi Formation and the Chhasra Formation [13]. The Khari Nadi Formation was deposited in the Aquitanian and the Chhasra Formation represents the Burdigalian [14]. The succession comprises a thick pile of sandstone, shale and siltstone with intervening thin layers of marl and limestone. These marl and limestone bands are richly fossiliferous and mainly yield varieties of benthic molluscs—bivalves and gastropods. *Persististrombus deperditus* is quite common at certain levels of this succession mainly from the uppermost part of the Khari Nadi Formation and lower part of the Chhasra Formation. In this study, out of several hundred specimens in our collection we have used only 72 sub-adult to adult specimens. These are chosen because of preservation of major parts of their apical and basal sides so that faithful measurements can be taken. Several of these specimens are preserved with shell. The specimens were collected randomly from three localities. The specimens come from the Khari Nadi Formation exposed near village Aida (Loc 1: 23°24′48.5″ N, 68°48′58″ E), and Chhasra Formation exposed about 2.5 km north of village

**Table 1.** Dimensions (in millimetre) of the specimens of *Persististrombus deperditus* (Sowerby, 1840) used in this study (PG/K/St: collection of strombid specimens from Kutch, Gujarat, western India deposited in the fossil collection of the Department of Geology, Presidency University, Kolkata, India).

| serial number | specimen number | diameter (mm) | height (mm) |
|---|---|---|---|
| macroconch | | | |
| 1 | PG/K/St 453 | 21.96 | 50.92 |
| 2 | PG/K/St 450 | 20.20 | 50.20 |
| 3 | PG/K/St 439 | 21.22 | 50.45 |
| 4 | PG/K/St 412 | 25.50 | 58.70 |
| 5 | PG/K/St 415 | 20.10 | 45.90 |
| 6 | PG/K/St 416 | 25.10 | 55.80 |
| 7 | PG/K/St 411 | 23.30 | 54.20 |
| 8 | PG/K/St 202 | 25.60 | 63.82 |
| 9 | PG/K/St 278 | 25.36 | 58.93 |
| 10 | PG/K/St 78 | 25.70 | 61.67 |
| 11 | PG/K/St 263 | 20.94 | 59.22 |
| 12 | PG/K/St 56 | 26.20 | 57.34 |
| 13 | PG/K/St 226 | 25.65 | 69.54 |
| 14 | PG/K/St 20 | 20.84 | 52.45 |
| 15 | PG/K/St 24 | 22.27 | 54.54 |
| 16 | PG/K/St 93 | 20.30 | 55.86 |
| 17 | PG/K/St 232 | 22.43 | 54.77 |
| 18 | PG/K/St 269 | 23.30 | 53.39 |
| 19 | PG/K/St 234 | 22.42 | 52.25 |
| 20 | PG/K/St 220 | 20.27 | 54.67 |
| 21 | PG/K/St 205 | 22.52 | 53.16 |
| 22 | PG/K/St 231 | 21.10 | 57.56 |
| 23 | PG/K/St 261 | 21.15 | 57.76 |
| 24 | PG/K/St 201 | 22.25 | 52.18 |
| 25 | PG/K/St 49 | 20.48 | 52.29 |
| 26 | PG/K/St 614 | 23.42 | 56.55 |
| 27 | PG/K/St 615 | 22.81 | 54.60 |
| 28 | PG/K/St 616 | 22.10 | 53.00 |
| 29 | PG/K/St 617 | 22.42 | 54.17 |
| 30 | PG/K/St 618 | 21.30 | 56.30 |
| 31 | PG/K/St 619 | 22.30 | 52.40 |
| 32 | PG/K/St 620 | 21.32 | 51.66 |
| 33 | PG/K/St 621 | 20.80 | 54.10 |
| 34 | PG/K/St 622 | 17.40 | 51.70 |
| 35 | PG/K/St 623 | 21.71 | 54.22 |
| 36 | PG/K/St 624 | 21.45 | 49.20 |
| 37 | PG/K/St 601 | 24.11 | 61.77 |
| 38 | PG/K/St 608 | 21.10 | 46.80 |
| 39 | PG/K/St 600 | 21.10 | 49.60 |

(*Continued.*)

| serial number | specimen number | diameter (mm) | height (mm) |
|---|---|---|---|
| 40 | PG/K/St 610 | 24.32 | 54.78 |
| 41 | PG/K/St 479 | 21.39 | 59.77 |
| 42 | PG/K/St 484 | 18.66 | 48.59 |
| 43 | PG/K/St 480 | 18.12 | 41.29 |
| 44 | PG/K/St 481 | 19.02 | 43.25 |
| 45 | PG/K/St 482 | 19.89 | 47.57 |
| 46 | PG/K/St 476 | 17.53 | 40.25 |
| 47 | PG/K/St 37 | 23.54 | 60.79 |
| 48 | PG/K/St 99 | 24.01 | 58.32 |
| 49 | PG/K/St 605 | 21.59 | 50.60 |
| 50 | PG/K/St 211 | 20.48 | 55.17 |
| 51 | PG/K/St 276 | 23.17 | 56.48 |
| 52 | PG/K/St 60 | 22.76 | 56.19 |
| 53 | PG/K/St 603 | 24.48 | 61.28 |
| microconch | | | |
| 54 | PG/K/St 7 | 15.11 | 37.05 |
| 55 | PG/K/St 17 | 15.47 | 38.37 |
| 56 | PG/K/St 124 | 16.15 | 39.55 |
| 57 | PG/K/St 431 | 16.15 | 39.41 |
| 58 | PG/K/St 6 | 17.96 | 37.24 |
| 59 | PG/K/St 435 | 15.83 | 34.77 |
| 60 | PG/K/St 433 | 14.67 | 38.22 |
| 61 | PG/K/St 477 | 14.61 | 40.91 |
| 62 | PG/K/St 473 | 14.32 | 42.54 |
| 63 | PG/K/St 478 | 16.60 | 40.55 |
| 64 | PG/K/St 9 | 15.15 | 36.35 |
| 65 | PG/K/St 475 | 14.84 | 37.56 |
| 66 | PG/K/St 474 | 15.55 | 36.28 |
| 67 | PG/K/St 472 | 15.60 | 40.63 |
| 68 | PG/K/St 11 | 16.65 | 38.47 |
| 69 | PG/K/St 470 | 18.19 | 40.78 |
| 70 | PG/K/St 436 | 13.36 | 32.54 |
| 71 | PG/K/St 625 | 16.76 | 37.40 |
| 72 | PG/K/St 626 | 15.31 | 40.84 |

Bhadra (Loc 2: 23°27′47.8″ N, 68°55′34″ E) and at Kankawati River near village Vinjan (Loc 3: 23°06′ N, 69°02′52″ E) (figure 2). All these specimens approach adulthood, which has been confirmed using one or more of the features discussed below.

## 3.2. Methodology

The dimension of a gastropod is traditionally represented by the shell height (figure 3), especially in the recent forms. In fossil strombids, either the apex or the anterior extremity of the siphonal canal or both are commonly broken. For this analysis, we have selected specimens with relatively less truncated ends to minimize error in the measurement of height. Still, measurement of the shell height (H) could have

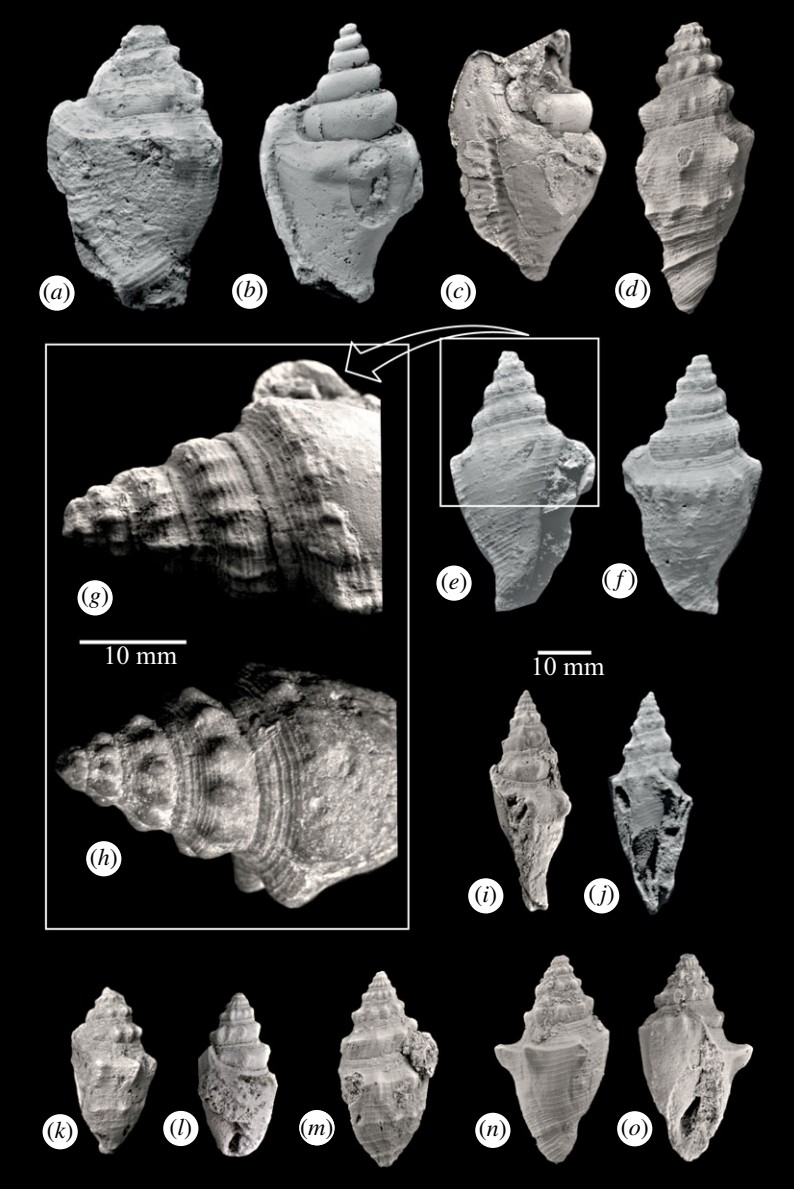

**Figure 1.** Photographs of macroconch (*a*–*h*) and microconch (*i*–*o*) specimens of *Persististrombus deperditus* (Sowerby, 1840) from the Miocene of Kutch, western India. (*a*) Abapertural view of PG/K/St 412. (*b*) Abapertural view of PG/K/St 232, an internal mould with adult apertural flaring and upturned suture. (*c*) Abapertural view of an internal mould PG/K/St 612 showing relatively long upturned suture and plications in the interior of outer lip. (*d*) Abapertural view of PG/K/St 37 showing strong shoulder tubercles and appearance of mid-whorl tubercles. (*e*–*g*) Apertural, abapertural and close-up views of PG/K/St 453. Close up (*g*) of the inset in (*e*) demonstrates weakening of shoulder tubercles. (*h*) Close up of PG/K/St 24 showing weakest shoulder tubercles preceded and followed by stronger ones. (*i*,*j*) Lateral and apertural views of PG/K/St 473. (*k*) Abapertural view of PG/K/St 6. Note mid-whorl tubercles in the last whorl. (*l*) Apertural view of PG/K/St 474, a microconch with upturned adult suture. (*m*) Abapertural view of PG/K/St 626. (*n*,*o*) Abapertural and apertural views, respectively, of PG/K/St 478 demonstrating sudden increase of strength of shoulder tubercles in the last whorl. Scale bars = 10 mm.

incorporated some error due to reconstruction. Hence, we have also measured the shell diameter (D) of the last whorl near the final position of aperture where the suture turns up adapically (figure 3). Measurements were taken using a digital slide caliper and used up to two decimal digits. Histograms were constructed with D and H data. The data distribution of designated macro- and microconch specimens were tested for normality by Shapiro–Wilk test ($p < 0.05$). Equality of mean, variance and distribution were tested using *t*-, *F*- and Kolmogorov–Smirnov tests, respectively ($p < 0.05$). Normal probability plots were also obtained for the data. We have performed discriminant analysis to get a quantitative evaluation of success of our visual classification into two size classes. All the statistical analyses and plots were obtained using the PAST 3.18 software [15].

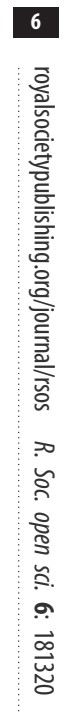

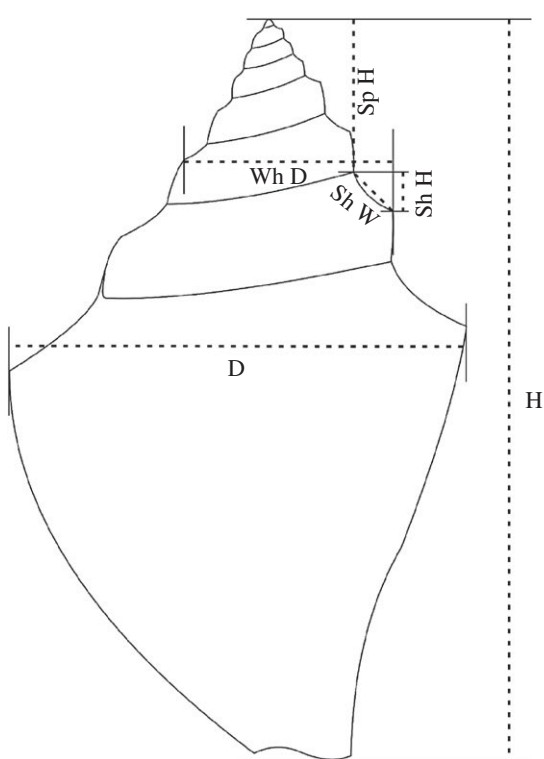

**Figure 2.** Geological map of the study area with the localities of collection shown (Loc 1–3).

**Figure 3.** Schematic diagram of an adult specimen of *P. deperditus* (Sowerby) showing dimensions measured for different analyses. See text for details.

A longitudinal data analysis has been carried out using 17 well-preserved specimens. Nine macroconch and eight microconch specimens have been used. Four parameters were measured consecutively at different ontogenetic stages. The stages are separated by 180° angular distance. The measured parameters are whorl diameter (Wh D), spire height (Sp H), shelf height (Sh H) and shelf

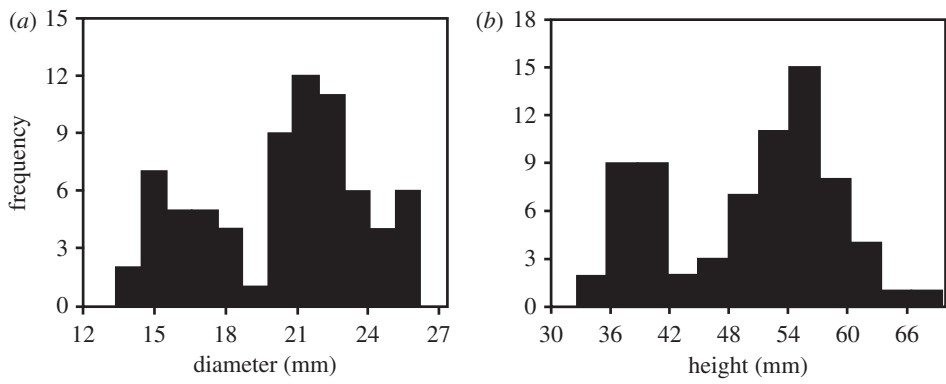

**Figure 4.** Histograms showing frequency distributions of shell diameter (*a*) and shell height (*b*) of *P. deperditus* (Sowerby).

width (Sh W) (figure 3). A maximum of nine and minimum of four stages could be measured in a specimen. Principal components analysis (PCA) was carried out on these data. The ratios between spire height and corresponding whorl diameter are plotted against Wh D.

Representative specimens were coated with magnesium oxide before they were photographed by Nikon D7000 DSLR camera (figure 1).

## 3.3. Determining adulthood

Adulthood in the family Strombidae is generally marked by an apical extension of the posterior canal and an upturned suture. Another feature commonly associated with the adult aperture in this family is a flared and thickened outer lip having plicate interior. In *Persististrombus deperditus*, apical extension of the suture is relatively short and rarely crosses the preceding whorl (figure 1*a,b,f,i,k,l*; however, 1*c* for an exception). Flared outer lip with plicate interior is preserved in only a few specimens of our collection (figure 1*b,c*). However, a few other features facilitated recognition of the final size here. Strength of the shoulder tubercles and surface ornamentation reduce considerably about one whorl before the final aperture (figure 1*g,h*). This area, which lies at the same level with the adult aperture, was in contact with the substrate in the living individual. Apparently, this reduction of ornamentation is an adaptation for smooth movement of the adult individual. This is also economical because anti-predatory ornaments were not required in this part. The strength of the shoulder tubercles abruptly increases after this stage (figure 1*g,h,k,n*). This increase is accompanied by sudden appearance of spirally arranged tubercles at the middle of the whorl side (figure 1*d,k,m*). We have restricted our analysis to specimens where one or more of these features could be studied.

## 4. Results

Visual inspection of our collection of *Persististrombus deperditus* from the Lower Miocene of Kutch indicated large variability in the adult shell size and the presence of broadly two size classes. The larger size class, referred as the macroconch, has shell diameter of about 20 mm and above, and shell height of 50 mm and above. The smaller size group, referred as the microconch, comprises of individuals with D mostly less than 15 mm, and H less than 40 mm (table 1). The histograms that have been prepared from the data of H and D broadly show two-peak distributions (figure 4). This supports the division of the assemblage into the macro- and the microconch. The Shapiro–Wilk test reveals that H and D for both the macro- and the microconch specimens are distributed normally ($p < 0.05$) (table 2). The normal probability plots arranged the data points for both the morphs largely along the reduced major axis regression lines, which were drawn for reference and comparison (figure 5). This also exhibits normal distribution of shell height and shell diameter for the two morphs. The probability of their means, variances and distributions being equal appears to be negligible ($p < 0.05$) (table 3). Because the variances of D and H for the two morphs can be significantly different, unequal variance *t*-test scores are also given in the table 3. However, *p*-values for both equal and unequal variance *t*-tests show significant difference between the macro- and the microconch. The discriminant analysis reveals that 95.83% of the specimens were correctly classified along the discriminant axis (figure 6). Only three of the macroconchs are classified as microconchs in the discriminant analysis. H has a higher loading (4.4091) on the

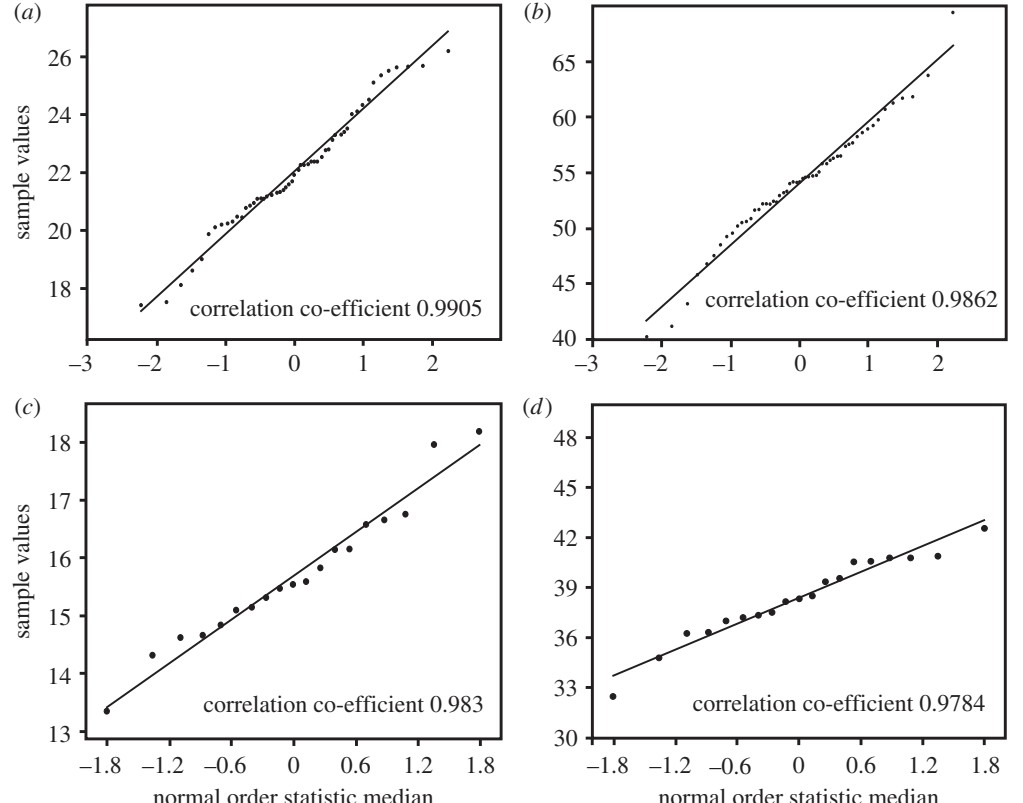

**Figure 5.** Normal probability plots with correlation co-efficient for shell diameter of macroconchs (*a*), shell height of macroconchs (*b*), shell diameter of microconchs (*c*), and shell height of microconchs (*d*) of *P. deperditus* (Sowerby). All the plots show strong correlation indicating normal distribution.

**Table 2.** Values of some important statistical parameters for *P. deperditus* (Sowerby) population used in this study.

| parameter | macroconch | | microconch | |
| --- | --- | --- | --- | --- |
| | H | D | H | D |
| Mean | 54.113 | 22.046 | 38.393 | 15.699 |
| Variance | 29.311 | 4.4168 | 6.0067 | 1.4346 |
| Shapiro−Wilk test *p* (normal) | 0.513 | 0.366 | 0.6245 | 0.7962 |

discriminant function than D (1.7803). The normal distribution, and significantly different means, variances and distributions indicate that the macro- and the microconch specimens represent natural populations, which differ significantly in their diameter and height.

The longitudinal data gathered from measurement of successive ontogenetic stages are given in table 4. PCA of the data and the plot of PC2 versus PC1 illustrate moderate degree of separation between the macro- and the microconch specimens along PC2 (figure 7*a*). PC1 accounts for more than 98% variability, whereas PC2 describes more than 1% (table 5). Distribution of loadings of the measured parameters demonstrates that PC1 is essentially the size axis, whereas PC2 is defined mainly by the ratio between Sp H and Wh D (table 5 and figure 7*b,c*). This ratio is plotted against Wh D (figure 8). This scatter plot indicates that the microconch is more slender than the macroconch.

## 5. Discussion

The analyses reveal significant difference in the adult size between the macro- and the microconch, which were separated by visual inspection. The microconch is significantly smaller and also more slender than

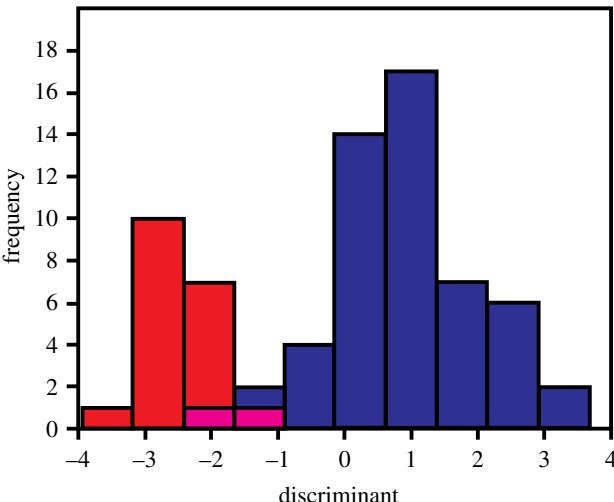

**Figure 6.** Histogram of discriminant projection values for the two morphs of *P. deperditus* (Sowerby). There is slight overlapping between the two morphs defined by visual examination.

**Table 3.** Probability values of two population tests for H and D data of the macro- and the microconch specimens of *P. deperditus* (Sowerby).

| test | H | D |
|---|---|---|
| *t-test* | | |
| *p* (same mean) | $5.9748 \times 10^{-19}$ | $2.2381 \times 10^{-19}$ |
| *p* (same mean, unequal variance) | $1.2558 \times 10^{-25}$ | $1.844 \times 10^{-22}$ |
| *F-test* | | |
| *p* (same variance) | 0.00056114 | 0.011006 |
| *Kolmogorov–Smirnov test* | | |
| *p* (same distribution) | $1.3665 \times 10^{-12}$ | $4.0553 \times 10^{-12}$ |

the macroconch. Size dimorphism in a gonochoristic species is generally attributed to differences due to sexes. However, intraspecific size dimorphism due to non-sexual causes are known. Such dimorphism is generally linked to ecophenotypic differences due to difference in external physico-chemical and biological conditions of the habitat such as temperature, latitude, moisture content, food availability, predation pressure, wave-current energy, etc. [16–21]. These factors commonly vary with geography. However, within-habitat variation in these factors is possible depending on local differences. Here, we examine whether sexual or non-sexual causes were responsible for the evolution of size dimorphism in *Persististrombus deperditus* by trying to refute them one by one.

## 5.1. Sexual dimorphism?

SSD in fossil mega-invertebrates is best exemplified and almost omnipresent in the Jurassic–Cretaceous ammonites (Ammonitida, Cephalopoda). There, the antidimorphs differ in size and other morphologies in adulthood, whereas they share juvenile features. They are accepted as sexual consorts if found from the same stratigraphic level and overlapping geographical areas [22,23]. The study of SSD in ammonites has a long history [24–27]. The phenomenon is now almost unequivocally accepted by ammonoid researchers in spite of the fact that nothing is known about anatomical differences in the soft part of the antidimorphs [23]. The macroconch is often twice to four times the microconch in size. However, overlapping size ranges of the macro- and the microconch are known [22]. They also generally differ in adult apertural modifications, ornamentation and some shape parameters. However, in contradiction to the above statement, dimorphic pair with identical coiling and ornamentation is also known [28]. The relative abundance of the macro- and the microconch also commonly differs and may be in the range of 100 to one in either way. The difference in numerical ratios between the

**Table 4.** Measurements (in millimetre) for longitudinal analysis. See text for details.

| specimen no. | macro-/microconch | Wh D | Sp H | Sh H | Sh W |
|---|---|---|---|---|---|
| PG/K/St 20 | macroconch | 20.43 | 17.87 | 5.71 | 6.74 |
| | | 17.29 | 13.96 | 4.21 | 4.76 |
| | | 13.4 | 11.89 | 3.08 | 3.31 |
| | | 11.13 | 8.57 | 2.89 | 3.31 |
| | | 8.81 | 6.69 | 2.1 | 2.2 |
| | | 7.37 | 5.62 | 1.77 | 1.69 |
| | | 6.19 | 4.57 | 1.24 | 1.36 |
| | | 4.6 | 3.7 | 0.92 | 1 |
| PG/K/St 24 | macroconch | 22.5 | 20.74 | 4.65 | 6.25 |
| | | 18.88 | 17.36 | 4.19 | 5.07 |
| | | 15.24 | 13.09 | 3.78 | 4.4 |
| | | 13.42 | 10.98 | 3.19 | 3.8 |
| | | 10.47 | 8.01 | 2.35 | 3.5 |
| | | 8.31 | 6.8 | 1.95 | 2.35 |
| | | 6.84 | 4.49 | 1.36 | 2.01 |
| | | 5.6 | 4.3 | 1.06 | 1.94 |
| PG/K/St 453 | macroconch | 20.77 | 21.47 | 3.88 | 4.87 |
| | | 18.1 | 17.89 | 3.56 | 4.36 |
| | | 14.81 | 13.62 | 2.91 | 3.81 |
| | | 12.24 | 11.31 | 2.7 | 2.94 |
| | | 9.89 | 8.68 | 2.04 | 2.35 |
| | | 8.43 | 7.32 | 1.6 | 2.06 |
| | | 6.67 | 4.96 | 1.11 | 1.34 |
| | | 5.18 | 4.12 | 0.71 | 0.99 |
| PG/K/St 412 | macroconch | 25.92 | 24.62 | 4.04 | 4.99 |
| | | 22.65 | 22 | 3.91 | 4.77 |
| | | 16.66 | 16.14 | 3.66 | 4.52 |
| | | 14.56 | 14.33 | 3.04 | 4.45 |
| | | 11.46 | 10.57 | 2.9 | 4.25 |
| | | 9.17 | 8.93 | 2.33 | 2.95 |
| PG/K/St 37 | macroconch | 22.09 | 23.97 | 5.98 | 7.34 |
| | | 16.83 | 19.16 | 5.4 | 5.88 |
| | | 15.5 | 16.09 | 5.12 | 5.38 |
| | | 13.83 | 12.04 | 3.45 | 4.18 |
| | | 10.72 | 9.69 | 3.09 | 3.63 |
| | | 9.36 | 7.62 | 2.59 | 2.77 |
| | | 7.72 | 6.51 | 2.04 | 2.24 |
| PG/K/St 479 | macroconch | 23.81 | 22.41 | 5.67 | 6.75 |
| | | 19.26 | 17.22 | 4.97 | 5.88 |
| | | 16.32 | 15.04 | 4.59 | 5.27 |
| | | 13.83 | 11.46 | 3.63 | 4.21 |
| | | 10.69 | 9.28 | 3.06 | 3.35 |

(*Continued.*)

| specimen no. | macro-/microconch | Wh D | Sp H | Sh H | Sh W |
|---|---|---|---|---|---|
| | | 8.88 | 6.14 | 2.23 | 2.65 |
| | | 6.89 | 5.68 | 1.79 | 2.34 |
| PG/K/St 415 | macroconch | 20.26 | 17.17 | 4.67 | 5.4 |
| | | 16.48 | 13.21 | 4.01 | 4.37 |
| | | 13.77 | 10.11 | 3.45 | 4.23 |
| | | 10.93 | 7.65 | 2.78 | 3.15 |
| | | 8.15 | 6.15 | 2.44 | 2.98 |
| | | 6.81 | 4.72 | 1.3 | 1.88 |
| | | 5.3 | 3.92 | 1.2 | 1.63 |
| | | 4.05 | 2.61 | 0.74 | 1.02 |
| PG/K/St 416 | macroconch | 24.48 | 22.3 | 5.97 | 6.99 |
| | | 19.31 | 19.12 | 5.76 | 6.34 |
| | | 16.07 | 14.71 | 4.04 | 4.82 |
| | | 14.06 | 12.69 | 3.92 | 4.09 |
| | | 11.26 | 9.47 | 3.26 | 3.67 |
| | | 10.04 | 8.83 | 2.09 | 2.6 |
| PG/K/St 439 | macroconch | 20.26 | 17.11 | 4.97 | 5.66 |
| | | 17.29 | 14.44 | 4.42 | 4.63 |
| | | 12.22 | 11.63 | 3.11 | 4.1 |
| | | 11.08 | 9.11 | 2.92 | 3.2 |
| | | 8.08 | 7.65 | 1.63 | 3.01 |
| PG/K/St 477 | microconch | 14.37 | 15.3 | 3.88 | 5.59 |
| | | 12.36 | 13.8 | 2.91 | 3.7 |
| | | 10.07 | 9.66 | 2.47 | 2.92 |
| | | 8.66 | 8.14 | 2.12 | 2.82 |
| | | 6.89 | 6.45 | 1.72 | 2.06 |
| | | 6.11 | 4.67 | 1.54 | 1.72 |
| | | 4.74 | 3.88 | 1.03 | 1.02 |
| | | 4.08 | 2.8 | 0.67 | 0.68 |
| PG/K/St 473 | microconch | 14.28 | 17.32 | 3.31 | 4.16 |
| | | 12.16 | 13.98 | 3.31 | 3.53 |
| | | 10.15 | 11.56 | 2.76 | 2.74 |
| | | 8.87 | 8.93 | 2.28 | 2.47 |
| | | 6.74 | 7.15 | 1.61 | 1.78 |
| | | 6.12 | 5.19 | 1.16 | 1.77 |
| | | 4.47 | 3.92 | 0.89 | 1.09 |
| | | 4.02 | 3.09 | 0.79 | 0.8 |
| PG/K/St 433 | microconch | 14.76 | 14.93 | 3.44 | 3.8 |
| | | 12.69 | 11.22 | 2.5 | 3.13 |
| | | 10.56 | 9.79 | 2.09 | 3 |
| | | 8.49 | 7.43 | 1.87 | 2.09 |
| | | 6.66 | 5.87 | 1.47 | 1.31 |

| specimen no. | macro-/microconch | Wh D | Sp H | Sh H | Sh W |
|---|---|---|---|---|---|
| | | 5.79 | 4.84 | 1.1 | 1.2 |
| | | 4.07 | 3.38 | 0.62 | 1.06 |
| | | 3.82 | 2.67 | 0.52 | 0.63 |
| PG/K/St 478 | microconch | 17.36 | 15.14 | 4.18 | 4.79 |
| | | 15 | 12.58 | 3.95 | 4.42 |
| | | 12.68 | 10.14 | 3.53 | 4.07 |
| | | 9.55 | 7.6 | 2.53 | 3.64 |
| | | 7.62 | 6.01 | 2.07 | 2.35 |
| | | 6.85 | 4.71 | 1.81 | 2.21 |
| | | 5.53 | 3.55 | 1.22 | 1.33 |
| | | 3.83 | 2.98 | 1.03 | 1.1 |
| PG/K/St 472 | microconch | 16.51 | 16.51 | 4.15 | 4.92 |
| | | 14.21 | 12.85 | 3.21 | 3.34 |
| | | 10.88 | 10.73 | 3.06 | 3.1 |
| | | 9.1 | 8.16 | 2.11 | 2.74 |
| | | 6.67 | 6.52 | 1.84 | 1.51 |
| | | 4.54 | 5.01 | 1.28 | 1.4 |
| PG/K/St 474 | microconch | 16.03 | 15.23 | 3.25 | 3.79 |
| | | 13.2 | 12.65 | 2.65 | 3.41 |
| | | 10.71 | 9.78 | 2.56 | 2.88 |
| | | 8.79 | 7.55 | 2.42 | 2.43 |
| | | 7.54 | 5.81 | 1.83 | 2.12 |
| | | 6.07 | 4.63 | 1.57 | 1.78 |
| | | 4.83 | 3.5 | 1.05 | 1.32 |
| | | 3.75 | 2.63 | 0.97 | 1.14 |
| PG/K/St 475 | microconch | 15.47 | 16.93 | 4.14 | 4.88 |
| | | 12.27 | 12.91 | 2.84 | 3.89 |
| | | 9.78 | 10.56 | 2.81 | 3.3 |
| | | 8.34 | 8.25 | 1.81 | 2.46 |
| | | 7 | 6.6 | 1.52 | 2.36 |
| | | 5.44 | 5.01 | 1.22 | 1.56 |
| | | 4.38 | 4.19 | 1 | 1.32 |
| | | 3.65 | 3.29 | 0.86 | 1.2 |
| PG/K/St 626 | microconch | 15.81 | 16.91 | 5.42 | 5.89 |
| | | 12.87 | 13.58 | 3.83 | 3.51 |
| | | 11.05 | 10.85 | 3.65 | 4.24 |
| | | 9.69 | 8.6 | 2.4 | 2.8 |
| | | 8.1 | 7.24 | 2.05 | 2.89 |
| | | 6.63 | 5.87 | 1.58 | 1.88 |
| | | 4.92 | 4.19 | 1.41 | 1.83 |

antidimorphs is generally attributed to facies and taphonomy [29]. The evolution and continuation of SSD in ammonoid lineages of the Jurassic and the Cretaceous appear to be phylogenetically linked [22,30]. Overwhelming and global presence of size dimorphism of similar nature in almost all the

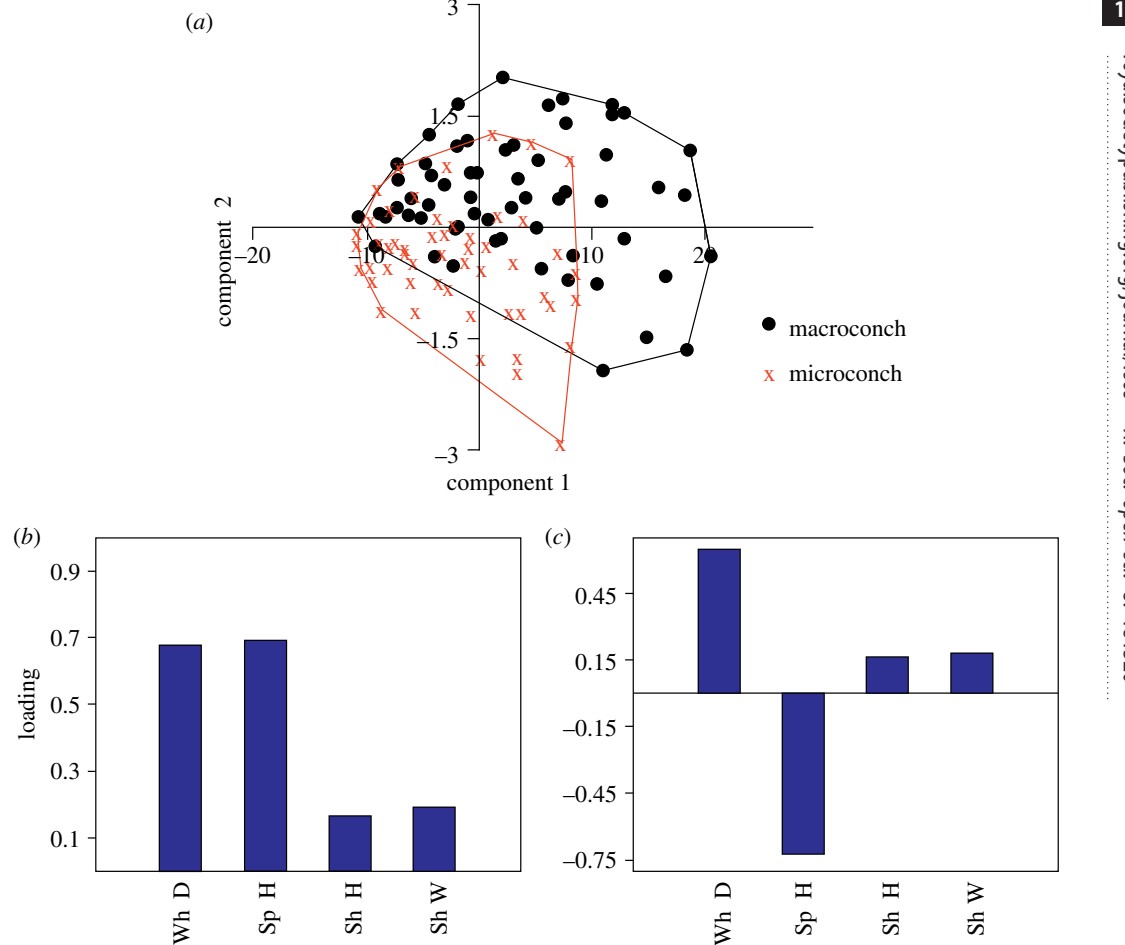

**Figure 7.** (*a*) Scatter plot from PCA on variance–covariance matrix in PC1–PC2 coordinate system. (*b,c*) Loading of measured parameters on PC1 (*b*) and PC2 (*c*), respectively.

**Table 5.** PCA scores (*a*) and loadings of each of the measured parameters on the principal components (*b*).

| PC | eigenvalue | % variance | | |
|---|---|---|---|---|
| (a) scores | | | | |
| 1 | 59.9779 | 98.057 | | |
| 2 | 0.736567 | 1.2042 | | |
| 3 | 0.40254 | 0.65811 | | |
| 4 | 0.049177 | 0.080399 | | |
| parameter | PC 1 | PC 2 | PC 3 | PC 4 |
| (b) loadings | | | | |
| Wh D | 0.67742 | 0.64673 | −0.35047 | 0.0034962 |
| Sp H | 0.69079 | −0.72302 | 0.0010819 | 0.0078693 |
| Sh H | 0.16452 | 0.16618 | 0.63201 | 0.73884 |
| Sh W | 0.19198 | 0.17712 | 0.69118 | −0.67383 |

Jurassic–Cretaceous ammonite lineages and its evolutionary sustenance led to the large-scale acceptance of its sexual origin among ammonite researchers.

SSD is present in all the living species of the family Strombidae wherever examined from this point of view. This fact prompted Cob *et al.* [9] to consider SSD the general condition of the lineage. SSD in living strombids has also been known for a long time [6–8]. In these living species, it is generally characterized primarily by moderate difference in size between the macro- and the microconch with overlapping in

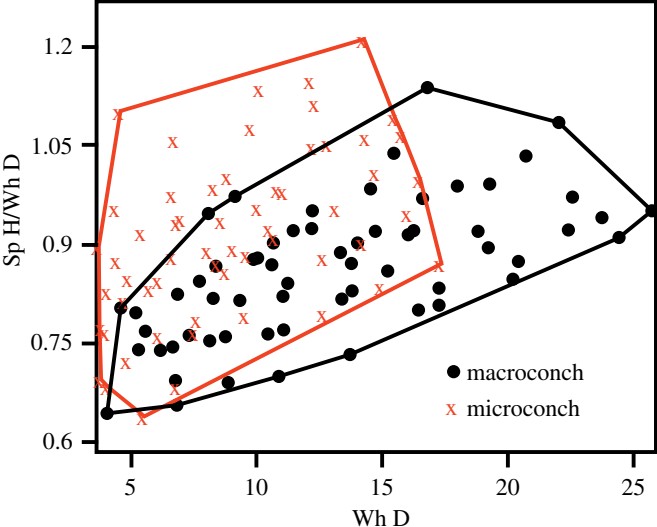

**Figure 8.** Sp H to Wh D ratios plotted against Wh D.

their size ranges [1,2,31]. They also differ in shape, with males having relatively slimmer shells than females [5]. Males are also known to have thinner lips than females [1]. Females are also more abundant than males [4,5]. Strombid SSD is essentially similar to that of ammonoids except that it is characterized by relatively smaller difference in size between the sexes. Significant difference in ornamentation has also not been reported in any of the dimorphic strombids.

SSD has never been reported in fossil strombids. *Persististrombus deperditus* resembles recent strombids in exhibiting moderate size dimorphism with some overlapping. Microconchs have slimmer shells than macroconchs. The lip thickness in most of the fossil specimens could not be properly studied and compared between the two varieties. The two morphs come from the same stratigraphic levels and from the same localities. Macroconchs are much more abundant than microconchs in all localities of collection. High abundance of the species wherever it was encountered, random orientation of its shells, low degree of fragmentation and abrasion, and presence of the specimens only in plane-bedded marl and limestone indicate that the fossils are autochthonous to parautochthonous. They apparently had substrate preference and flourished in abundance whenever substrate conditions were suitable. Independent determination of sex is not possible in *P. deperditus* as is possible in the living representatives of Strombidae. However, given the identical nature of dimorphism in it with those in living strombids, sexual origin of the size dimorphism in *P. deperditus* is difficult to refute. This also supports evolutionary sustenance of the identical SSD in living strombids.

## 5.2. Ecophenotypic difference?

Closely related species sometimes show distinct geographical pattern of morphological variation, which is commonly guided by latitude [18]. Intra- and interspecific size variations are known to occur in many animals, including molluscs, due to difference in ecological conditions of the habitats. However, significant size difference has also been reported between closely related species and between sub-populations of a species, which are separated by only about 10 m distance [18,19,21]. Johannesson *et al.* [19] reported wide variation in the size of *Littorina saxatilis* (Olivi, 1792) (Littorinidae, Gastropoda) population from Galicia. It is a direct developer species having indeterminate growth. It lives in the intertidal zone. They found large difference in size between individuals living in the lower shore and those in the upper shore [19]. There is a steep environmental gradient between the lower and the upper shores in terms of temperature, salinity, subaerial exposure, substrate, topography, wave energy, predation pressure and several other factors ([21] and references therein). These differences are reflected in large differences in the shape and the size of species and sub-populations that thrive in these spatially close areas [18,21]. Faunal diversity is generally low in the intertidal areas because of strong fluctuation in ecological factors. But, there are a number of examples that record differences—including in size—between sub-populations living in sympatry in the intertidal habitat [18,19,21]. However, prominent non-sexual size difference between sub-populations living in sympatry are not known from other marine habitats.

Molluscs that lack a planktotrophic dispersive larval ontogeny are known to display more prominent differences between their populations in terms of size than those with planktotrophic larval ontogeny

([19] and references therein). However, strombids with planktotrophic larval development are also known to vary significantly in size. Savazzi [2] observed that conspecific strombids living in the intertidal zone are much smaller than those living in the subtidal part a few tens of metres away. The intertidal and the subtidal areas differ markedly with respect to hydrodynamic condition, substrate type, temperature variation, food availability and predation pressure. Within the same strombid population, however, size varies only between 20 and 30% [2].

The specimens of *Persististrombus deperditus* from Kutch were mostly collected from limestone beds near the top of the Khari Nadi Formation and the basal part of the Chhasra Formation. These are biostromal composite shell concentrations ([32] and references therein). The shells belong to a diverse benthic community that colonized repeatedly. The community flourished when siliciclastic supply was poor [32,33]. Lower part of the Khari Nadi Formation is dominated by fine sandstone and siltstone. These were deposited in the intertidal zone and are largely barren. With transgression and deepening, the basin became starved of clastic sediment supply. This allowed colonization of this relatively quiet and stable shelf by varieties of benthic animals. The shells suffered only minor within-habitat transportation [32]. *Persististrombus deperditus* is one of the common members of this community. They are not encountered in the lower part of the Khari Nadi Formation, which represents intertidal zone.

In summary, distinct intraspecific non-sexual size difference generally occurs (1) between sub-populations that live in upper and lower shores of the intertidal zone or (2) between populations that thrive in distinct habitats separated by some barrier—small or large. *Persististrombus deperditus* flourished in the Kutch basin only when transgression curtailed silisiclastic supply to the deepened shelf. The population belonged to a community that lived in a stable habitat, which was not segmented into distinct zones based on difference in ecological factors. Hence, the size difference between the micro- and the macroconch is difficult to explain by ecophenotypic causes.

It appears that large size variation in *Persististrombus deperditus* as ecophenotypic difference is more easily refuted than as sexual difference. Hence, difference in sexes is considered the most plausible reason for the development of size dimorphism in *P. deperditus*. Savazzi's [2] observation that fossil strombids of the same assemblage may display a size ratio of 4 : 1 can also include SSD.

## 5.3. Evolution of SSD in *Persististrombus deperditus*

The family Strombidae perhaps appeared in the Upper Cretaceous but became a common component of benthic mollusc faunas in shallow tropical and sub-tropical seas only from the later part of the Palaeogene [2,3]. The present report indicates that SSD, comparable to that of the living strombids, had appeared quite early in the history of this lineage. Another stromboid species belonging to the genus *Tibia* Röding, 1798 (Rostellariidae, Stromboidea, Gastropoda) co-occurs with *Persististrombus deperditus* in the Lower Miocene of Kutch. It also shows large variability in adult size in spite of a similar number of whorls in the adulthood, indicating the presence of possible SSD (K Halder & S Paira 2018, unpublished data).

In all examined living species of Strombidae, SSD is characterized by larger adult females [1,4,5,9,34]. Cob *et al*. [9] observed that while size difference between the sexes in *Laevistrombus canarium* is small, most of the sampled large individuals are females and the small individuals are males. In the four recent strombid species from the USA, live-caught specimens of which were used for captive breeding experiment, the female shells were observed to be higher and heavier than the male shells although their ranges overlap to a great extent (see table 2 in [31]). In *P. deperditus* we have found significant difference between the size of the macro- and the microconch although there is overlapping in their size ranges. Wherever sexual consorts of a living mollusc species are observed to have prominent difference in their adult size, the females are usually larger [35]. Fossil molluscs including ammonoids [22,23] and nautiloids [36–38] followed the same pattern of SSD. Hence, the macroconch of *P. deperditus* has greater probability of being the female.

It has been observed that the macroconch of *P. deperditus* is considerably more abundant than the microconch. Cob *et al*. [4] also observed abundance of females over males throughout the year of study in *Laevistrombus canarium* from Malaysia (see also [5]). Disparate sex ratios are a commonplace observation in sexually dimorphic species of ammonites, where it has been seen to vary from nearly one to 100 in either way [29]. Disparate sex ratio has also been observed in nautiloids [37,38]. In these cephalopods, the disparity has been attributed to seasonal segregation of the sexes or taphonomic bias [22,23,29,39]. Seasonal niche partitioning between the sexes is also known in living cephalopods [40]. It has been documented that in living strombids, sexes sometimes segregate during burial behaviour [41]. This may be a part of their reproductive behaviour pattern ([42] and references therein). At the present

state of knowledge, it is uncertain whether the relative abundance of the morphs of *P. deperditus* is real or due to niche separation.

## 5.4. Why dimorphism: general remarks

SSD can evolve in an organism in response to one or more of the following causes.

### 5.4.1. Sexual selection

A male can father more offspring than a female can bear by mating with more than one female. This increases the reproductive success of the male but a female's reproductive success does not increase by mating with multiple males. This ensues in selection pressure on males. In response to this selection pressure males either fight among themselves to win over a female or mate a female coercively. As a consequence larger males are generally selected because strength is the key factor in this selection [43,44]. Larger males are also commonly selected in response to this selection pressure where epigamic choice by females is involved. There may be other responses to this selection pressure that result in smaller males. A smaller male with higher agility and manoeuvrability is selected because of greater success in search of mate, especially in fluid media [45,46]. Smaller males that mature early can spend maximum energy in reproduction [47]. Such progenetic growth of the male due to reproductive haste is common in molluscs [40,48].

### 5.4.2. Fecundity selection

The female grows to a larger size than the male and often differs in body shape to physically accommodate a large clutch of eggs. SSD in some bivalves and gastropods has been attributed to this factor [48–50].

### 5.4.3. Niche partitioning

If males and females occupy different ecological niches—either geographically or based on resource utilization—occupation of the eco-space increases and intersexual competition decreases [43,51]. As a consequence, sexes adapt and evolve independently. Such dimorphism is reported in molluscs. The dimorphism produced in such cases is generally seen to be characterized by larger females [52,53]. Niche partitioning itself does not explain the larger size of females. This may be in response to some other factor, such as dietary habits [54].

### 5.4.4. Non-adaptive causes

SSD can result as a by-product of change in some other character, e.g. feeding rates in juveniles or paedomorphic growth in one sex relative to the other [55]. It is worth mentioning that slower feeding rate and progenetic growth with early maturity in the male had been reported and explained in adaptive terms also [56].

## 5.5. Possible causes of SSD in *Persististrombus deperditus*

Sexual selection that selects larger males due to male–male fighting or involvement of coercion in copulation is not known in any mollusc or any other invertebrate organism. Epigamic choice by females is also primarily a phenomenon known in more complex groups of vertebrates. In the recent Strombidae, often several males including those that belong to different species are seen to attempt copulation with a female individual simultaneously [57,58]. A male attempting copulation with another male has also been observed [57,58]. Epigamic choice in such a group is almost impossible. Small size would not provide much agility and manoeuvrability to a benthic crawler, like a strombid gastropod. Although early maturity in males is common in molluscs, progenetic growth has not been reported in males of living Strombidea. Instead, neotenic growth of the male with respect to the female has been observed in *Laevistrombus canarium* [4]. Possibility of reproductive haste as the driving mechanism for evolution of SSD in *P. deperditus* seems low.

The larger and wider shell of the female gastropod to accommodate more eggs definitely provides higher success in producing large number of progeny. Hence, fecundity selection can be a probable cause of SSD in strombids. Cob *et al*. [9] also suggested that females of *Laevistrombus canarium*

allocated more energy to gonad production. Similar observations were made in other strombids and also other gastropods ([5,35] and references therein).

Ecological niche separation between males and females leading to independent adaptation and evolution of the sexes can be a possible cause behind SSD in *P. deperditus*. This can also explain the significantly discrepant relative abundance of the morphs. However, information on living strombids is inadequate to support this hypothesis unequivocally. Living strombids are gregarious, and herbivorous or detritivorous in habit [2]. They commonly show a burial behaviour, during which sometimes sexes segregate [41]. But, this separation does not adequately explain the size difference.

Evolution of SSD as a fabricational noise (*sensu* [59]), such as by paedomorphic growth of one sex compared to the other, can only be established if such modification does not involve any adaptive advantage. In the case of living strombids with SSD the role of growth process could not be dissociated from that of adaptation.

It appears from the above discussion that fecundity selection is the most plausible cause for the evolution of SSD in *Persististrombus deperditus*. The only other factor that might have played some role was niche separation between the sexes.

# 6. Conclusion

SSD is present in all the living species of Strombidae wherever examined but was unknown in fossil strombids. The oldest SSD in the gastropod family Strombidae is reported here in *Persististrombus deperditus* from the Lower Miocene of Kutch, western India. The two morphs are distinguished mainly by the adult size, which is represented in this analysis by the shell diameter and the shell height. The SSD seems to have served the purpose of accommodating a larger clutch of eggs in the larger female. Discrepant relative abundance of the sexes, if not actual, might have resulted from niche separation between the two sexes. Appearance of SSD in such an early stage of evolution of the family suggests that further investigation may reveal more dimorphic species in this lineage.

Data accessibility. The specimens used in this study are deposited in the fossil collection of the Department of Geology, Presidency University, Kolkata, India. Measurement data used here are provided in the tables within the text. All the statistical analyses were done using the PAST 3.18 software [15].

Authors' contribution. Both the authors were involved in the collection and preparation of specimens. K.H. conceived and designed the study, coordinated the analyses, interpreted the results, and wrote and revised the manuscript. S.P. participated in acquisition and handling of data, conducted statistical analyses and drafted the manuscript. Both the authors gave final approval for publication.

Competing interests. We have no competing interests.

Funding. The first author received partial financial assistance from the Science and Engineering Research Board, Department of Science and Technology (Project nos. SR/S4/ES-653/2012 and EMR/2016/002583), Government of India.

Acknowledgements. The authors are grateful to A. Beu, H. Parent and G. Kronenberg, who reviewed an older version of the manuscript and provided valuable suggestions. Suggestion of three anonymous reviewers are also acknowledged.

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
