## [Reviewer comments · Royal Society Open Science]

Review History

RSOS-171755.R0 (Original submission)

Review form: Reviewer 1

Is the manuscript scientifically sound in its present form?

No

Are the interpretations and conclusions justified by the results?

No

Is the language acceptable?

No

Is it clear how to access all supporting data?

Yes

Do you have any ethical concerns with this paper?

No

Have you any concerns about statistical analyses in this paper?

No

Recommendation?

Reject

Comments to the Author(s)

This ms aims to describe sexual size dimorphism (SSD) in Indian fossil strombids from the Miocene. Unfortunately, the authors have no means of rejecting alternative hypotheses regarding why the two size classes they observe exist. As such, the ms fails to meet the standards for publication, and I recommend that it be rejected.

In order to suggest SSD, one must be able to discriminate between the sexes independently of shell size, quantify some size difference(s) within the sample, and finally show that the size differences discriminates between the sexes in the same manner as the independent data. The authors identified a bimodal size distribution within their sample, but have no independent, corroborating data that conclusively can assign the sex to individuals. Their assertion that "all females are bigger, so all bigger snails are female" may indeed be true, but with no way to assign sex, their data doesn't mean much. Also, there are other explanations for the observed bimodal distribution. Micro- and macrohabitat differences, large and small genotypes, and selection due to predation have all been shown to affect phenotype presence and distribution in gastropod populations. The authors did not test any other hypothesis that could explain their data.

Since I recommend rejection, I did not go line by line to correct the grammar in the ms. However, the authors are encouraged to have their English more thoroughly edited in any future submissions/resubmissions of this work.

Review form: Reviewer 2 (Shiladri Das)

Is the manuscript scientifically sound in its present form?

Yes

Are the interpretations and conclusions justified by the results?

Yes

Is the language acceptable?

Yes

Is it clear how to access all supporting data?

Yes

Do you have any ethical concerns with this paper?

No

Have you any concerns about statistical analyses in this paper?

No

Recommendation?

Accept with minor revision (please list in comments)

Comments to the Author(s)

In 'Introduction' part the full form of 'SSD' already mentioned. Please replace 'sexual size dimorphism' by 'SSD' from rest of the manuscript.

Author should provide better photographs in Figure 1 for (a), (l), (n) and (o).

Review form: Reviewer 3 (Silvina Van Der Molen)

Is the manuscript scientifically sound in its present form?

No

Are the interpretations and conclusions justified by the results?

No

Is the language acceptable?

No

Is it clear how to access all supporting data?

Yes

Do you have any ethical concerns with this paper?

No

Have you any concerns about statistical analyses in this paper?

No

Recommendation?

Accept with minor revision (please list in comments)

Comments to the Author(s)

This article reports the first record of sexual size dimorphism in fossil Strombidae and its possible evolutionary implications. Overall I think that this is a simple and tidy paper that discusses interesting hypotheses about size dimorphism. In my opinion the paper fit the journal scope and is quite well structured. I recommend the publication on the Royal Society Open Science with revisions.

The specific comments are attached in the manuscript (Appendix A).

General comments:

- I suggest that the authors get editing help from someone with full proficiency in English.
- The manuscript (including figures and tables) needs format editing according to the journal instructions.
- The use of some categorical adjectives to describe the results is a bit speculative (e.g. conclusively, in all probabilities, etc.).
- The way the authors refer to the overlapping between morphs and intermediate sizes is confusing.

Decision letter (RSOS-171755.R0)

23-Feb-2018

Dear Dr Halder:

Manuscript ID RSOS-171755 entitled "First record of sexual size dimorphism in fossil Strombidae (Gastropoda) from the Miocene of Kutch, western India and its evolutionary implications" which you submitted to Royal Society Open Science, has been reviewed. The comments from reviewers are included at the bottom of this letter.

In view of the criticisms of the reviewers, the manuscript has been rejected in its current form. However, a new manuscript may be submitted which takes into consideration these comments.

The reviews are very divergent, making it difficult to arrive at a final decision. However, I am swayed by the fact that in the absence of a clear means of establishing sex of fossil taxa, it is difficult to establish sexual dimorphism. This is a valid criticism, but one common to most studies of fossil organisms. It is something that you should address explicitly in a revised version of your manuscript, specifically to avoid accusation of circularity. All reviewers suggest some improvement of use of English is called for, and this should also be given priority when revising your manuscript.

Please note that resubmitting your manuscript does not guarantee eventual acceptance, and that your resubmission will be subject to peer review before a decision is made.

Your resubmitted manuscript should be submitted by 23-Aug-2018. If you are unable to submit by this date please contact the Editorial Office.

Please note that Royal Society Open Science will introduce article processing charges for all new submissions received from 1 January 2018. Charges will also apply to papers transferred to Royal Society Open Science from other Royal Society Publishing journals, as well as papers submitted as part of our collaboration with the Royal Society of Chemistry (<http://rsos.royalsocietypublishing.org/chemistry>). If your manuscript is submitted and accepted for publication after 1 Jan 2018, you will be asked to pay the article processing charge, unless you request a waiver and this is approved by Royal Society Publishing. You can find out more about the charges at <http://rsos.royalsocietypublishing.org/page/charges>. Should you have any queries, please contact openscience@royalsociety.org.

on behalf of Jon Blundy (Subject Editor)
openscience@royalsociety.org

Reviewers' Comments to Author:

Reviewer: 1

Comments to the Author(s)

This ms aims to describe sexual size dimorphism (SSD) in Indian fossil strombids from the Miocene. Unfortunately, the authors have no means of rejecting alternative hypotheses regarding why the two size classes they observe exist. As such, the ms fails to meet the standards for publication, and I recommend that it be rejected.

In order to suggest SSD, one must be able to discriminate between the sexes independently of shell size, quantify some size difference(s) within the sample, and finally show that the size differences discriminates between the sexes in the same manner as the independent data. The authors identified a bimodal size distribution within their sample, but have no independent, corroborating data that conclusively can assign the sex to individuals. Their assertion that "all females are bigger, so all bigger snails are female" may indeed be true, but with no way to assign sex, their data doesn't mean much. Also, there are other explanations for the observed bimodal distribution. Micro- and macrohabitat differences, large and small genotypes, and selection due to predation have all been shown to affect phenotype presence and distribution in gastropod populations. The authors did not test any other hypothesis that could explain their data.

Since I recommend rejection, I did not go line by line to correct the grammar in the ms. However, the authors are encouraged to have their English more thoroughly edited in any future submissions/resubmissions of this work.

Reviewer: 2

Comments to the Author(s)

In 'Introduction' part the full form of 'SSD' already mentioned. Please replace 'sexual size dimorphism' by 'SSD' from rest of the manuscript.

Author should provide better photographs in Figure 1 for (a), (l), (n) and (o).

Reviewer: 3

Comments to the Author(s)

This article reports the first record of sexual size dimorphism in fossil Strombidae and its possible evolutionary implications. Overall I think that this is a simple and tidy paper that discusses interesting hypotheses about size dimorphism. In my opinion the paper fit the journal scope and is quite well structured. I recommend the publication on the Royal Society Open Science with revisions.

The specific comments are attached in the manuscript.

General comments:

- I suggest that the authors get editing help from someone with full proficiency in English.

- The manuscript (including figures and tables) needs format editing according to the journal instructions.

- The use of some categorical adjectives to describe the results is a bit speculative (e.g. conclusively, in all probabilities, etc.).
- The way the authors refer to the overlapping between morphs and intermediate sizes is confusing.

Author's Response to Decision Letter for (RSOS-171755.R0)

See Appendix B.

RSOS-181320.R0

Review form: Reviewer 2 (Shiladri Das)

Is the manuscript scientifically sound in its present form?

Yes

Are the interpretations and conclusions justified by the results?

Yes

Is the language acceptable?

Yes

Is it clear how to access all supporting data?

Yes

Do you have any ethical concerns with this paper?

No

Have you any concerns about statistical analyses in this paper?

Yes

Recommendation?

Accept as is

Comments to the Author(s)

Thank you for submitting a well formatted manuscript.

Review form: Reviewer 3 (Silvina Van Der Molen)

Is the manuscript scientifically sound in its present form?

Yes

Are the interpretations and conclusions justified by the results?

Yes

Is the language acceptable?

Yes

Is it clear how to access all supporting data?

Not Applicable

Do you have any ethical concerns with this paper?

No

Have you any concerns about statistical analyses in this paper?

No

Recommendation?

Accept as is

Comments to the Author(s)

Dear authors,

I consider that the manuscript has improved enough to be published.

Decision letter (RSOS-181320.R0)

18-Jan-2019

Dear Dr Halder,

I am pleased to inform you that your manuscript entitled "First record of sexual size dimorphism in fossil Strombidae (Gastropoda) from the Miocene of Kutch, western India and its evolutionary implications" is now accepted for publication in Royal Society Open Science.

You have the opportunity to archive your accepted, unbranded manuscript, but access to the full text must be embargoed until publication.

Articles are normally press released. For this to be effective we set an embargo on news coverage corresponding to the publication date of the article. We request that news media and the authors do not publish stories ahead of this embargo (when final version of the article is available).

on behalf of Prof Jon Blundy (Subject Editor)
openscience@royalsociety.org

Reviewer comments to Author:
Reviewer: 2

Comments to the Author(s)
Thank you for submitting a well formatted manuscript.

Reviewer: 3

Comments to the Author(s)
Dear authors,
I consider that the manuscript has improved enough to be published.

Appendix A

ROYAL SOCIETY OPEN SCIENCE

First record of sexual size dimorphism in fossil Strombidae (Gastropoda) from the Miocene of Kutch, western India and its evolutionary implications

Journal:	Royal Society Open Science
Manuscript ID	RSOS-171755
Article Type:	Research
Date Submitted by the Author:	02-Nov-2017
Complete List of Authors:	Halder, Kalyan; Presidency University, Geology Paira, Somnath; Presidency University, Geology
Subject:	Palaeontology < EARTH SCIENCES
Keywords:	Sexual size dimorphism, Strombidae, Kutch, western India, Miocene, fecundity selection
Subject Category:	Earth science

**First record of sexual size dimorphism in fossil**
**Strombidae (Mollusca, Gastropoda) from the**
**Miocene of Kutch, western India and its**
**evolutionary implications**

**Kalyan Halder and Somnath Paira**

*Department of Geology, Presidency University, 86/1, College Street, Kolkata – 700073, West*
*Bengal, India.*

**Corresponding author**

**Kalyan Halder**

**e-mail: kalyan.geol@presiuniv.ac.in**

**Keywords:** Sexual size dimorphism, Strombidae, Kutch, western India, Miocene, fecundity selection.

**1. Summary**

*Persististrombus deperditus* (Sowerby) from the lower Miocene of Kutch, Gujarat, western India is
represented by two size classes in our collection. Simple statistical analyses separate the size morphs
conclusively. The size difference presumably resulted from sexual dimorphism where the larger
morph is the female. All the living species of the family Strombidae, wherever examined, are
characterized by similar sexual size dimorphism (SSD). Other possible causes of large size variation
within a species generally occur in different populations or subpopulations. This is the first report of
SSD in a fossil strombid gastropod. Longitudinal analysis of diameter data indicates possible
involvement of positive allometry in the ontogeny of both the morphs, and neoteny in the growth of
the microconch with respect to the macroconch. It is argued that fecundity selection was the driving
force behind the evolution of SSD in this gonochoristic gastropod species.

2. Introduction

The family Strombidae is present in the Holocene exclusively in shallow tropical and sub-tropical seas. Wherever strombid species are found they are generally quite abundant [1]. The family is characterized by determinate growth. Once the shell attains final size it undergoes substantial change in apertural features, such as upturned suture, often flared and thickened outer lip that sometimes bears spines. Final shell size and shape are attained in this family before sexual maturity [2, 3].

All the living species of Strombidae are gonochoristic. All the examined species of the family exhibit sexual dimorphism with respect to size where females are larger than males ([4] and references therein). Sexual size dimorphism in some recent species of Strombidae has been known for a long time (e.g., [5-7]), and Cob *et al.* [8] considered it the general characteristic of the genus "*Strombus*". Size ranges of the two sexes often overlap, but members in the larger size end are generally females whereas those in the smaller size end are males (e.g., [8]). In spite of the determinate growth and apparent ease of recognition of adulthood because of the presence of adult apertural modifications, sexual dimorphism has never been reported in fossil Strombidae. Dependence of palaeontological studies solely on hard part morphologies and presence of overlapping size ranges between the sexes might have discouraged such documentation. Here we report sexual size dimorphism in *Persististrombus deperditus* (Sowerby, 1840) from the Miocene of Kutch, western India. We also seek to understand the possible mechanism of development of dimorphism and causes of its evolution.

Abbreviations

SSD: sexual size dimorphism; PG/K/St: collection of strombid specimens from Kutch, Gujarat, western India deposited in the fossil collection of the Department of Geology, Presidency University, Kolkata, India; D: shell diameter; H: shell height.

3. Dimorphic *Persististrombus deperditus* (Sowerby, 1840)

Strombus deperditus Sowerby, 1840 is known from the lower Miocene of Kutch and Pakistan [9, 10]. Harzhauser *et al.* [11] included it (misspelled as *deperditus*) in the genus *Persististrombus* Kronenberg & Lee, 2007 and put *Strombus nodosus* Sowerby, 1840 into its synonymy. The latter comes from the same stratigraphic level and geographic locality as it. The species is abundant in Kutch. Harzhauser *et al.* [11] provided a formal systematic description of the species.

The species is fusiform, anomphalous, dextrally coiled with elevated and step like spire. The spire is shorter than the last whorl. The last whorl overlaps about 70% of the preceding whorl. The base is conical and extended into a short and slightly curved anterior siphonal canal. An angular shoulder separates a moderately wide and slightly sloping shelf from the rather flat whorl side. Its surface bears characteristic ornament having prominent spiral ribbing and shoulder tubercles. The species is easily identifiable because of these prominent morphological features. Interestingly, it occurs in a wide range of adult sizes in Kutch. On visual inspection they can be broadly grouped in two size classes, which do not differ significantly in other features. The number of whorls in all the adult specimens is about eight. The larger and the smaller size groups are referred here as the macro- and the microconch respectively (table 1, figure 1). 
The height of the largest macroconch is twice that of the smallest microconch although overlapping is present in the size ranges of the two groups. Therefore, it is difficult to assign some specimens to either of the groups. Hence, we have performed simple statistical analyses to see the viability of these groups.

4. Material and methods

4.1. Material studied

The lower Miocene is represented in Kutch, Gujarat by the Khari Nadi Formation and the Chhasra Formation [12]. The Khari Nadi Formation was deposited in the Aquitanian and the Chhasra Formation represents the Burdigalian [13]. The succession is comprised of a thick pile of sandstone, shale and siltstone with intervening thin layers of marl and limestone. These marl and limestone bands are richly fossiliferous and mainly yield varieties of benthic molluscs – bivalves and gastropods. *Persististrombus deperditus* is quite common at certain levels through this succession. In this study, out of several hundred specimens in our collection we have used only 72 sub-adult to adult specimens. These are chosen because of preservation of major parts of their apical and basal sides so that faithful measurements can be taken. Many of these specimens are preserved with shell. The specimens were collected randomly from three localities. The specimens come from the Khari Nadi Formation exposed near village Aida (Loc 1: 23°24' 48.5" N, 68°48' 58" E), and Chhasra Formation exposed about 2.5 km north of village Bhadra (Loc 2: 23°27' 47.8" N, 68°55' 34" E) and at Kankawati River near village Vinjan (Loc 3: 23°06' N, 69°02' 52" E) (Fig. 2). All these specimens approach adulthood, which has been confirmed using one or more of the features discussed below.

4.2. Methodology

The dimension of a gastropod is traditionally represented by the shell height (figure 3), especially in the recent forms. In fossil strombids either the apex or the anterior extremity of the siphonal canal or both are commonly broken. For this analysis we have selected specimens with relatively less truncated ends to minimize error in the measurement of height. Still, measurement of the shell height (H) could have incorporated some error due to reconstruction. Hence, we have also measured the shell diameter (D) of the last whorl near the final position of aperture where the suture turns up adapically (figure 3). Measurements were taken using a digital slide caliper and used up to two decimal digits (table 1). Histograms were constructed with D and H data (figure 4). The data distribution of designated macro- and microconch specimens were tested for normality by Shapiro-Wilk test ($p < 0.05$) (table 2). Equality of mean, variance and distribution were tested using t-, F- and

Kolmogorov-Smirnov tests respectively ($p < 0.05$) (table 3). Normal probability plots were also
obtained for the data (figure 5). All the statistical analyses and plots were obtained using the PAST
3.12 software [14].

A longitudinal data analysis has been carried out using six well-preserved specimens where a set of
consecutive measurements of diameter at different ontogenetic stages were possible. Three
macroconch and three microconch specimens have been used. Eight diameter data have been
measured in each of the specimens. First measurement was that of the last whorl (D), which was used
in the previous analyses. Successively other measurements corresponding to earlier ontogenetic
stages were taken at 180° angular distance apart (figure 3). These eight diameter data (table 4) for each
of the specimens were plotted in a univariate space keeping early ontogenetic stage to the left (figure
6).

Representative specimens were coated with magnesium oxide before they were photographed by
Nikon D7000 DSLR and Sony RX10 III cameras (figure 1).

4.3. Determining adulthood

Adulthood in the family Strombidae is generally marked by an apical extension of the posterior canal
and an upturned suture. Other features commonly associated with the adult aperture in this family
are a flared and thickened outer lip having plicate interior. In *Persististrombus deperditus* apical
extension of the suture is relatively short and rarely crosses the preceding whorl (figure
1(b),(g),(l),(m),(s) etc., however, 1(e) for an exception). Flared outer lip with plicate interior is
preserved in only a few specimens of our collection (figure 1(b),(e)). However, here a few other
features facilitated recognition of the final size. Nearly one whorl before the final aperture strength of
the shoulder tubercles and surface ornamentation reduce considerably (figs 1(c),(i),(j)). This area,
which lies at the same level with the adult aperture, was in contact with the substrate in the living
individual. Apparently, this reduction of ornamentation is an adaptation for smooth movement of the
adult individual. This is also economical because anti-predatory ornaments were not required in this

part. The strength of the shoulder tubercles abruptly increases after this stage (figure
1(d),(f),(j),(n),(o),(q)). This increase is accompanied by sudden appearance of spirally arranged
tubercles at the middle of the whorl side (figure 1(d),(f),(n)-(p)). We have restricted our analysis only
to specimens where one or more of these features could be studied. Measurements were taken only
when adapical extension of the suture is observed.

16 17 18 19 20 21 22 23 24 25 26 27 28 29 30 31 32 33 34 35 36 37 38 39 40 41 42 43 44 45 46 47 48 49 50 51 52 53 54

Visual inspection of our collection of *Persististrombus deperditus* from the lower Miocene of Kutch
indicated large variability in the adult shell size and the presence of broadly two size classes. The
larger size class, referred as the macroconch, has the shell diameter of about 20 mm and above, and
the shell height of 50 mm and above. The smaller size group, referred as the microconch, comprises of
individuals with D mostly less than 15 mm, and H less than 40 mm. The histograms that have been
prepared from the data of H and D broadly show two-peak distributions, thus separating the
macroconch from the microconch (figure 4). The Shapiro–Wilk test reveals that H and D for both the
macro- and the microconch specimens are distributed normally ($p < 0.05$) (table 2). The normal
probability plots arranged the data points for both the morphs largely along the reduced major axis
regression lines, which were drawn for reference and comparison (figure 5). This also indicates
normal distribution of shell height and shell diameter for the two morphs. The probability of their
means, variances and distributions being equal appears to be negligible ($p < 0.05$) (table 3). Because the
variances of D and H for the two morphs can be significantly different unequal variance t-test scores
are also given in the Table 3. However, p-values for both equal and unequal variance t-tests show
significant difference between the macro- and the microconch. Normal distribution, and significantly
different means, variances and distributions indicate that the macro- and the microconch specimens
represent natural populations, which differ significantly in their final diameter and height.

In the longitudinal data analysis, all the three selected macroconch specimens were plotted above the
three microconch specimens throughout the measured ontogenetic stages. All the curves that joined

the eight ontogenetic stages of each of the six specimens are slightly concave up indicating possible
increase in the rate of growth of the diameter during ontogeny, i.e., involvement of possible
allometry. The macroconch specimens further show an acceleration of growth of the diameter over
the microconch specimens from stage four, thereby making the curve more acutely concave and
separating the macroconch group from the microconch more clearly in later ontogeny (figure 6).
There is a tendency of flattening of the curves from stage seven, which demarcates approaching
adulthood.

6. Discussion

6.1. Evolution of sexual size dimorphism (SSD) in *Persististrombus deperditus*

All the analyses reveal significant difference in the final size between the macro- and the microconch,
which were separated by visual inspection. Size dimorphism in a gonochoristic species is generally
attributed to differences due to sexes. The size dimorphism in *Persististrombus deperditus* can be
considered as sexual in nature in view of the fact that SSD is present in all the living species of the
family wherever examined from this point of view. This fact prompted Cob *et al.* [8] to consider SSD
the general condition of the lineage. In these living species it is also characterised primarily by
difference in size between the macro- and the microconch. SSD, while known from ancient
gonochoristic prosobranch gastropods [15], is not as prevalent as in the Mesozoic ammonites
(Ammonitida, Cephalopoda) [16], and has never been reported in fossil strombids.

Intra-specific size variation is known to occur in many animals including molluscs due to other 47 causes. They include difference in physical conditions of the habitat or micro-habitat, e.g.,
temperature, latitude, moisture content, food availability, predation pressure, water energy etc. [17-
20]. While such size variations generally occur between geographically isolated populations of a
species significant size difference has also been reported between sub-populations separated by only
about 10m distance [19]. Johannesson *et al.* [19] reported wide variation in size in *Littorina saxatilis*
(Olivi, 1792) (Littorinidae, Gastropoda) population from Galicia. It is a direct developer species

having indeterminate growth that lives in the intertidal zone. They found large difference in size
between individuals living in lower shore and those in upper shore zones [19]. Molluscs lacking
planktotrophic dispersive larval ontogeny are known to display more prominent differences between
their populations in terms of size ([19] and references therein). However, strombids with
planktotrophic larval development are also known to vary significantly in size. Savazzi [2] observed
that conspecific strombids living in the intertidal zone are much smaller than those living in the
subtidal part a few tens of meters away. Within the same population, however, size varies between
20% and 30% [2].

The macro- and the microconch specimens of the present study were found to occur together in all
the sites of collection. High abundance of the species wherever it was encountered, random
orientation of its shells, presence of the specimens only in plane bedded marl and limestone indicate
that the fossils are largely autochthonous to parautochthonous. They apparently had substrate
preference and flourished in abundance whenever substrate conditions were suitable. Co-occurrence
of the two morphs in autochthony indicates that they belong to the same population. It appears that
two distinct size classes in the same population can best be explained by sexual size dimorphism.
Hence, difference in sexes appears to be the most likely reason for the development of the size
dimorphism in *Persististrombus deperditus*. Savazzi's [2] observation that fossil strombids of the same
assemblage may display a size ratio of 4:1 can also include SSD.

The family Strombidae perhaps appeared in the Upper Cretaceous but became a common component
of benthic mollusc faunas in shallow tropical and sub-tropical seas only since the later part of the
Paleogene [2,3]. The present report indicates that SSD, comparable to that of the living strombids, had
appeared quite early in the history of this lineage. Another stromboid species belonging to the genus
*Tibia* Röding, 1798 (Rostellariidae, Stromboidea, Gastropoda) co-occurs with *Persististrombus*
*deperditus* in the lower Miocene of Kutch. It also shows large variability in adult size in spite of similar
number of whorls in the adulthood, indicating presence of possible SSD (Halder and Paira
unpublished data).

It appears from the longitudinal data analysis that the size difference between the macro- and the
microconch arose by a heterochronic process where the microconch is neotenic with respect to the
macroconch. It may be mentioned that the macro- and the microconch have similar number of whorls
and the measurements were taken at comparable growth stages with reference to their adulthood.
Hence, the neoteny that is involved in the growth of diameter of the microconch is in terms of
comparable growth stages and not necessarily with respect to age. It may further be noted here that
the heterochronic term 'neoteny' is used in a relative sense and does not indicate any evolutionary
direction. It is the same as 'acceleration' of the macroconch with respect to the growth of the
microconch.

Cob *et al.* [4] recorded allometry in the ontogeny of the living *Laevistrombus canarium* (Linnaeus, 1758)
from Malaysia and difference in growth rate between the sexual variants of this species. At the age of
one year the males of the species attain a height of 48.54 mm whereas the females attain 54.54 mm.
The males of the species mature at a smaller size, and they also live slightly longer than the females
[4].

In all examined living species of Strombidae SSD is characterised by larger adult females [1,4,8,21].
Cob *et al.* [8] observed that while size difference between the sexes in *Laevistrombus canarium* is small
most of the sampled large individuals are females and small individuals are males. In the four recent
strombid species from the USA, live caught specimens of which were used for captive breeding
experiment, female shells were observed to be higher and heavier than male shells although their
ranges overlap to a great extent (see table 2 in [22]). In *Persististrombus deperditus* we have found
significant difference between the sizes of macro- and microconch although there is overlapping in
their size ranges. Wherever sexual consorts of a living mollusc species are observed to have
prominent difference in adult sizes, the females are usually larger [23]. Fossil molluscs including
strombids, in all likelihood, followed the same pattern of SSD. Hence, the macroconch of *P. deperditus*
represents, in all probability, the females.

It may be pertinent to mention that Cob *et al.* [8] reported imposex variety of *Laevistrombus canarium*,
which is the largest and males the smallest. Imposex is generally developed due to presence of
pollutant tributyltin (TBT) compounds [24]. TBT induced imposex has also been reported in strombid
species *Lobatus gigas* (Linnaeus, 1758) [25]. However, imposex has also been reported from gastropod
samples collected before the use of TBT [26]. But, majority consensus is still inclined towards a strong
correlation between TBT and imposex. It may be mentioned that trimorphism has also been reported
in the Mesozoic ammonites [16]. In species where size ranges of the two sexes differ relatively less
and have overlapping, shell character based studies often tend to distinguish an intermediate size
variety.  In *Persististrombus deperditus* an intermediate size variety may be separated by visual 22
inspection and validated by some statistical means. But establishing such a biological disorder like
imposex is impossible in this nearly 20 million year old gastropod species. Further, anthropogenic
pollutants like TBT were absent in the seas of that time.

It has been observed that the macroconch of *Persististrombus deperditus* is considerably more abundant
than the microconch. Cob *et al.* [4] also observed abundance of females over males throughout the
35 year of study in *Laevistrombus canarium* from Malaysia. Disparate sex ratios are a commonplace
observation in sexually dimorphic species of ammonites, where it has been seen to vary from nearly
one to 100 in either way. There, the disparity has been attributed to seasonal segregation of the sexes
or taphonomic bias [16,27,28]. Seasonal niche partitioning between the sexes is also known in living
cephalopods [29]. It has been documented that in living strombids sexes sometimes segregate during
burial behaviour [30], which may be a part of their reproductive behaviour pattern ([31] and
references therein). At the present state of knowledge it is uncertain whether the relative abundance
of the morphs of *P. deperditus* is real.

**6.2. Why dimorphism - general remarks**

SSD can evolve in an organism in response to one or more of the following causes.

*Sexual selection.* A male can father more offspring than a female can bear by mating with more than
one female, thereby increasing its reproductive success but a female's reproductive success does not
increase by mating with multiple males. This ensues in selection pressure on males. In response to
this selection pressure males either fight among themselves to win over a female or mate a female
coercively. As a consequence larger males are generally selected because strength is the key factor in
this selection [32,33]. Larger males are also commonly selected in response to this selection pressure
where epigamic choice by females is involved. There may be other responses to this selection pressure
that result in smaller males. A smaller male with higher agility and maneuverability is selected
because of greater success in search of mate, especially in fluid media [34,35]. Further, smaller males
that mature early can spend maximum energy in reproduction [36]. Such progenetic growth of the
male due to reproductive haste is common in molluscs [29,37].

*Fecundity selection.* Females grow to a larger size than males and often differ in body shape to
physically accommodate a large clutch of eggs. SSD in some bivalves and gastropods has been
attributed to this factor [37-39].

*Niche partitioning.* If males and females occupy different ecological niches - either geographically or
based on resource utilisation - occupation of the eco-space increases and intersexual competition
decreases [32,40]. As a consequence, sexes adapt and evolve independently. Dimorphism produced in
such cases is generally seen to be characterised by larger females. 39

*Non-adaptive causes.* SSD can result as a by-product of change in some other character, e.g., feeding
rates in juveniles or paedomorphic growth in one sex relative to the other [41]. It is worth mentioning
that slower feeding rate and progenetic growth with early maturity in male had been reported and
explained in adaptive terms also [42].

**6.3. Possible causes of SSD in *Persististrombus deperditus***

Sexual selection that selects larger males due to male-male fighting or involvement of coercion in
copulation is not known in any mollusc or any other invertebrate organism. Epigamic choice by
females is also primarily a phenomenon known in more complex groups of vertebrates. In recent
Strombidae, often several males including those that belong to different species are seen to attempt
copulation with a female individual simultaneously [43,44]. A male attempting copulation with
another male has also been observed [43,44]. Epigamic choice in such a group is almost impossible.
Small size would not provide much agility and maneuverability to a benthic crawler, like a strombid
gastropod. Although early maturity in males is common in molluscs, progenetic growth has not been
reported in males of living Strombidea. Instead, neotenic growth of the male with respect to the
female has been observed in *Laevistrombus canarium* [4]. The Kutch species from the Miocene
demonstrated similar possibility of neoteny of the microconch in terms of growth stages. If this also
means neoteny with respect to absolute age, then “reproductive haste” cannot be forwarded as a
driving mechanism of SSD in this case.

The larger size of the female gastropod to accommodate more eggs definitely provides higher success
in producing large number of progenies and hence fecundity selection can be a probable cause of SSD
in strombids. Cob *et al.* [8] also suggested that females of *Laevistrombus canarium* allocated more
energy to gonad production.

Ecological niche separation between males and females leading to independent adaptation and
evolution of the sexes can be a possible cause behind SSD in *Persististrombus deperditus* because it
separates the resource utilisation between the sexes of the species, thereby lowering intersexual
competition. This can also explain the significantly discrepant relative abundance of the morphs.
However, information on living strombids is inadequate to support this hypothesis unequivocally.
Living strombids are gregarious, and herbivorous or detritivorous in habit [2]. They commonly show
a burial behavior, during which sometimes sexes segregate [30]. But, this separation does not
adequately explain the size difference.

Evolution of SSD as a fabricational noise (*sensu* [45]), such as by paedomorphic growth of one sex
compared to the other, can only be established if such modification does not involve any adaptive
advantage. We, here, found possible involvement of a heterochronic mechanism in the evolution of
SSD in the Miocene strombid species from Kutch. But this phenomenon cannot be dissociated from
adaptive causes.

It appears from the above discussion that fecundity selection is the most plausible cause for the
evolution of SSD in *Persististrombus deperditus*. The only other factor that might have played some role
was niche separation between the sexes.

24 25 7. Conclusions 26 27

Sexual size dimorphism is present in all the living species of Strombidae wherever examined but was
unknown in fossil strombids. The oldest SSD in the gastropod family Strombidae is reported here in
*Persististrombus deperditus* from the lower Miocene of Kutch, western India. The two morphs are
distinguished mainly in the adult size, which is represented in this analysis by the shell diameter and
the shell height. The intersexual difference in the shell diameter appears to have evolved by a
heterochronic mechanism involving rate of growth leading to narrower microconchs. The SSD seems
to have served the purpose of accommodating larger clutch of eggs in the larger female. Discrepant
abundance of the sexes, if not actual, might have resulted from niche separation between the two
sexes. Appearance of SSD in such early stage of evolution of the family suggests that further
investigation may reveal more dimorphic species in this lineage.

**Acknowledgements**

The authors are grateful to A. Beu, H. Parent and G. Kronenberg, who reviewed an older version of
the manuscript and provided valuable suggestions.

Funding Statement

The first author received partial financial assistance from the Department of Science and Technology (Project no. SR/S4/ES-653/2012), Government of India.

Data Accessibility

The specimens used in this study are deposited in the fossil collection of the Department of Geology, Presidency University, Kolkata, India. Measurement data used here are provided in the tables within the text.

All the statistical analyses were done using the PAST 3.12 software (Hammer *et al.* 2001)

Competing interests

We have no competing interests.

Authors' Contribution

Both the authors were involved in the collection and preparation of specimens. KH conceived and designed the study, co-ordinated the analyses, interpreted the results, and wrote and revised the manuscript. SP participated in acquisition and handling of data, conducted statistical analyses, and drafted the manuscript. Both the authors gave final approval for publication.

References

[revised manuscript text omitted]

**Table captions**

**Table 1.**

Dimensions (in mm) of the specimens of *Persististrombus deperditus* (Sowerby, 1840) used in this study.

**Table 2.**

Values of some important statistical parameters for *P. deperditus* (Sowerby) population used in this study.

**Table 3.**

Probability values of two population tests for H and D data of the macro- and microconch specimens of *P.*
*deperditus* (Sowerby).

**Table 4.**

Successive measurements of diameter (in mm) at eight positions of the specimens of *P. deperditus* (Sowerby)
used in the longitudinal analysis of this study. Each measurement is 180° apart from its preceding, as shown in
Fig. 3.

**Figure captions**

**Figure 1.**

Photographs of macroconch ((a)–(k)) and microconch ((l)–(t)) specimens of *Persististrombus deperditus*
(Sowerby, 1840) from the Miocene of Kutch, western India. (a) abapertural view of PG/St 412. (b)
abapertural view of PG/St 232, an internal mould with adult apertural flaring and upturned suture.
(c–d) apertural and abapertural views respectively of PG/St 484. Note reduced strength of shoulder
tubercles in the last whorl in apertural view (c). (e) abapertural view of an internal mould PG/St 612
showing relatively long upturned suture and plications in the interior of outer lip. (f) abapertural
view of PG/St 37 showing strong shoulder tubercles and appearance of mid-whorl tubercles. (g–i)
abapertural, apertural and close up views of PG/St 453 respectively. Close up (i) of the inset in (h)
demonstrates weakening of shoulder tubercles. (j) close up of PG/St 24 showing weakest shoulder
tubercles preceded and followed by stronger ones. (k) abapertural view of PG/St 482. (l) apertural

view of PG/St 431. (m) apertural view of PG/St 474, a microconch with upturned adult suture. (n)
abapertural view of PG/St 6. Note mid-whorl tubercles in the last whorl. (o) abapertural view of
PG/St 11. (p) abapertural view of PG/St 626. (q-r) abapertural and apertural views respectively of
PG/St 478 demonstrating sudden increase of strength of shoulder tubercles in the last whorl. (s-t)
lateral and apertural views of PG/St 473 respectively. Scale bars=10mm.

**Figure 2.**

Geological map of the study area with the localities of collection shown (Loc 1-3).

**Figure 3.**

Schematic diagram of an adult specimen of *P. deperditus* (Sowerby) showing dimensions measured for
different analyses. PQ and HI represent shell height (H) and shell diameter (D) respectively.

Successive measurements of diameter at different ontogenetic stages for the longitudinal analysis
were taken along AB, BC, CD, DE, EF, FG, GH and HI.

**Figure 4.**

Histograms showing frequency distributions of shell diameter (a) and shell height (b) of *P. deperditus*
(Sowerby).

**Figure 5.**

Normal probability plots with correlation co-efficient for shell diameter of macroconchs (a), shell
height of macroconchs (b), shell diameter of microconchs (c), and shell height of microconchs (d) of *P.*
*deperditus* (Sowerby). All the plots show strong correlation indicating normal distribution.

**Figure 6.**

Univariate plot of successive diameter measurements at different ontogenetic stages of the six
specimens of *P. deperditus* (Sowerby) - three macroconchs (PG/St/20, 24 and 453) and three
microconchs (PG/St/477, 473 and 433) - used in the longitudinal analysis. Microconchs show slower
(neotenic) rate of growth with respect to macroconchs.

Table 1. Dimensions (in mm) of the specimens of *Persististrombus deperditus* (Sowerby, 1840) used in this study.

Serial number	Specimen number	Diameter (mm)	Height (mm)
Macroconch			
PG/K/St 453	21.96	50.92
PG/K/St 450	20.20	50.20
PG/K/St 439	21.22	50.45
PG/K/St 412	25.50	58.70
PG/K/St 415	20.10	45.90
PG/K/St 416	25.10	55.80
PG/K/St 411	23.30	54.20
PG/K/St 202	25.60	63.82
PG/K/St 278	25.36	58.93
PG/K/St 78	25.70	61.67
PG/K/St 263	20.94	59.22
PG/K/St 56	26.20	57.34
PG/K/St 226	25.65	69.54
PG/K/St 20	20.84	52.45
PG/K/St 24	22.27	54.54
PG/K/St 93	20.30	55.86
PG/K/St 232	22.43	54.77
PG/K/St 269	23.30	53.39
PG/K/St 234	22.42	52.25
PG/K/St 220	20.27	54.67
PG/K/St 205	22.52	53.16
PG/K/St 231	21.10	57.56
PG/K/St 261	21.15	57.76
PG/K/St 201	22.25	52.18
PG/K/St 49	20.48	52.29
PG/K/St 614	23.42	56.55
PG/K/St 615	22.81	54.60
PG/K/St 616	22.10	53.00
PG/K/St 617	22.42	54.17
PG/K/St 618	21.30	56.30
PG/K/St 619	22.30	52.40
PG/K/St 620	21.32	51.66
PG/K/St 621	20.80	54.10
PG/K/St 622	17.40	51.70
PG/K/St 623	21.71	54.22
PG/K/St 624	21.45	49.20
PG/K/St 601	24.11	61.77
PG/K/St 608	21.10	46.80
PG/K/St 600	21.10	49.60
PG/K/St 610	24.32	54.78
PG/K/St 479	21.39	59.77
PG/K/St 484	18.66	48.59

PG/K/St 480	18.12	41.29
PG/K/St 481	19.02	43.25
PG/K/St 482	19.89	47.57
PG/K/St 476	17.53	40.25
PG/K/St 37	23.54	60.79
PG/K/St 99	24.01	58.32
PG/K/St 605	21.59	50.60
PG/K/St 211	20.48	55.17
PG/K/St 276	23.17	56.48
PG/K/St 60	22.76	56.19
PG/K/St 603	24.48	61.28
Microconch			
PG/K/St 7	15.11	37.05
PG/K/St 17	15.47	38.37
PG/K/St 124	16.15	39.55
PG/K/St 431	16.15	39.41
PG/K/St 6	17.96	37.24
PG/K/St 435	15.83	34.77
PG/K/St 433	14.67	38.22
PG/K/St 477	14.61	40.91
PG/K/St 473	14.32	42.54
PG/K/St 478	16.60	40.55
PG/K/St 9	15.15	36.35
PG/K/St 475	14.84	37.56
PG/K/St 474	15.55	36.28
PG/K/St 472	15.60	40.63
PG/K/St 11	16.65	38.47
PG/K/St 470	18.19	40.78
PG/K/St 436	13.36	32.54
PG/K/St 625	16.76	37.40
PG/K/St 626	15.31	40.84

Table 2. Values of some important statistical parameters for *P. deperditus* (Sowerby) population used in this study.

Parameter	Macroconch		Microconch	
	H	D	H	D
Mean	54.113	22.046	38.393	15.699
Variance	29.311	4.4168	6.0067	1.4346
Shapiro-Wilk test				
P (normal)	0.513	0.366	0.6245	0.7962

Table 3. Probability values of two population tests for H and D data of the macro- and microconch specimens of *P. deperditus* (Sowerby).

Test	H	D
t test		
p (same mean)	5.9748E ⁻¹⁹	2.2381 E ⁻¹⁹
p (same mean, unequal variance)	1.2558E ⁻²⁵	1.844E ⁻²²
F test		
p (same variance)	0.00056114	0.011006
Kolmogorov-Smirnov test		
p (same distribution)	1.3665E ⁻¹²	4.0553E ⁻¹²

Table 4. Successive measurements of diameter (in mm) at eight positions of the specimens of *P. deperditus* (Sowerby) used in the longitudinal analysis of this study. Each measurement is 180° apart from its preceding, as shown in Fig. 3.

	Specimen	AB	BC	CD	DE	EF	FG	GH	HI
Macroconch	PG/St/20	5.75	7.29	8.55	10.5	12.2	14.85	18.15	20.84
	PG/St/24	5.47	6.45	8.6	10.4	12.95	15.8	19.82	22.27
	PG/St/453	5.65	7.07	8.85	9.85	13.23	16.51	20.09	21.96
Microconch	PG/St/477	3.89	4.56	5.06	6.53	8.76	10.02	12.83	14.61
	PG/St/473	3.91	4.5	6.05	7.2	8.5	10.31	12.45	14.32
	PG/St/433	3.88	4.37	5.62	6.98	7.88	9.88	11.7	14.67

10mm

Appendix B

Reviewers' Comments to Author:

Reviewer: 1

Comments to the Author(s)

(1) This ms aims to describe sexual size dimorphism (SSD) in Indian fossil strombids from the Miocene. Unfortunately, **the authors have no means of rejecting alternative hypotheses regarding why the two size classes they observe exist**. As such, the ms fails to meet the standards for publication, and I recommend that it be rejected.

(2) In order to suggest SSD, **one must be able to discriminate between the sexes independently of shell size**, quantify some size difference(s) within the sample, and finally show that the size differences discriminates between the sexes in the same manner as the independent data. The authors identified a bimodal size distribution within their sample, but **have no independent, corroborating data that conclusively can assign the sex to individuals**. Their assertion that "all females are bigger, so all bigger snails are female" may indeed be true, but with no way to assign sex, their data doesn't mean much. Also, **there are other explanations for the observed bimodal distribution**. Micro- and macrohabitat differences, large and small genotypes, and selection due to predation have all been shown to affect phenotype presence and distribution in gastropod populations. The authors did not test any other hypothesis that could explain their data.

(3) Since I recommend rejection, I did not go line by line to correct the grammar in the ms. However, the authors are encouraged to **have their English more thoroughly edited** in any future submissions/resubmissions of this work.

Authors' response: (1) We had provided a brief discussion in the initial submission regarding rejection of alternative hypotheses. However, it was brief. In this revised version we have elaborated on it.

(2) Assignment of sex to individuals by some independent means is possible in the domain of biology, but not in palaeontology. In the revised MS we have specifically addressed this issue and taken the approach of refuting hypotheses concerning sexual and asexual origin of the dimorphism. Asexual factors that can create size difference between subpopulations leaving in sympatry, especially in marine molluscs, have been reviewed. It has been found that the asexual factors are more easily refuted than the sexual cause in the case of this strombid species.

(3) We have given care to English grammar and edited the language of the MS.

Reviewer: 2

Comments to the Author(s)

(1) In 'Introduction' part the full form of 'SSD' already mentioned. Please replace 'sexual size dimorphism' by 'SSD' from rest of the manuscript.

(2) Author should provide better photographs in Figure 1 for (a), (l), (n) and (o).

Authors' response: (1) Retained the full form (sexual size dimorphism) only in the first mention under a 1st order heading.

(2) Some of the photographs have been deleted. However, we feel, all the photographs mentioned by the reviewer are not that bad. They might look somewhat poor in the PDF.

Reviewer: 3

Comments to the Author(s)

(1) This article reports the first record of sexual size dimorphism in fossil Strombidae and its possible evolutionary implications. **Overall I think that this is a simple and tidy paper that discusses interesting hypotheses about size dimorphism.** In my opinion the paper fit the journal scope and **is quite well structured.** I recommend the publication on the Royal Society Open Science with revisions.

The specific comments are attached in the manuscript.

General comments:

-(2) I suggest that the authors get editing help from someone with full proficiency in English.

-(3) The manuscript (including figures and tables) needs format editing according to the journal instructions.

-(4) The use of some categorical adjectives to describe the results is a bit speculative (e.g. conclusively, in all probabilities, etc.).

-(5) The way the authors refer to the overlapping between morphs and intermediate sizes is confusing.

Authors' response: (1) Thank you.

(2) We have made necessary editing.

(3) Made necessary changes in the figures and tables following instructions and recent online papers published in RSOS.

(4) Made necessary changes.

(5) We agree. Deleted that part.